# New Insights in Assessing AKI 3 Risk Factors and Predictors Associated with On-Pump Surgical Aortic Valve Replacement

**DOI:** 10.3390/diagnostics15172211

**Published:** 2025-08-30

**Authors:** Anca Drăgan, Adrian Ştefan Drăgan

**Affiliations:** 1Department of Cardiovascular Anaesthesiology and Intensive Care, “Prof. Dr. C.C. Iliescu” Emergency Institute for Cardiovascular Diseases, 258 Fundeni Road, 022328 Bucharest, Romania; 2Faculty of General Medicine, Carol Davila University of Medicine and Pharmacy, 8 Eroii Sanitari Boulevard, 050474 Bucharest, Romania; adrian-stefan.dragan2023@stud.umfcd.ro

**Keywords:** acute kidney injury, risk factors, cardiac surgery, inflammatory indexes, sex differences

## Abstract

**Background**: Acute kidney injury (AKI) following cardiac surgery can lead to chronic kidney disease, increased hospitalization costs, and higher mortality risk. Our retrospective study identified risk factors of severe AKI (AKI 3) in patients undergoing on-pump surgical aortic valve replacement (SAVR). Additionally, we analyzed the significance of inflammatory indexes and risk scores in predicting AKI 3, focusing on sex differences. These findings could provide cost-efficient tools for clinical practice to identify patients at risk, improve preoperative risk stratification, and personalize monitoring. **Methods**: We reviewed the on-pump SAVR patients from our tertiary center between 2022 and 2024. **Results**: Out of 422 patients, 121 (28.67%) experienced AKI, including 27 (6.39%) AKI 3 patients. The multivariable binary logistic regression identified AKI 3 independent risk factors: hemostasis reintervention (OR9.76, CI 95%: 3.565–26.716, *p* = 0.001), early postoperative vasoactive-inotropic score (VIS) (OR1.049, CI 95%: 1.013–1.086, *p* = 0.007), postoperative lymphocyte (OR2.252, CI 95%: 1.224–4.144, *p* = 0.009). Preoperative systemic inflammatory response index (AUC0.700, *p* = 0.019), preoperative aggregate index of systemic inflammation (AUC0.712, *p* = 0.011), postoperative platelet-to-lymphocyte ratio (PLR) (AUC 0.759, *p* = 0.001), and the delta value of preoperative-to-postoperative PLR (AUC0.752, *p* = 0.001) were better predictors of AKI 3 occurrence in female SAVR patients than the additive EuroSCORE (AUC0.692, *p* = 0.011), but were less accurate compared to EuroSCORE II (AUC0.841, *p* = 0.001). None of the studied inflammatory indexes or additive EuroSCORE predicted our endpoint in male SAVR patients, while Thakar score was able to predict it exclusively in males. **Conclusions**: Early postoperative VIS, lymphocyte count, and hemostasis reintervention were independent risk factors for severe AKI in SAVR patients. There is a differentiation between males and females from the AKI prediction perspective.

## 1. Introduction

Acute kidney injury (AKI) remains one of the most significant postoperative complications following cardiac surgery, affecting 5–43% patients [1]. The pathophysiology of AKI in this context is multifactorial. It is mainly influenced by several factors, including reduced kidney functional reserve, hemodynamic disturbances such as renal hypoperfusion and venous congestion, exposure to nephrotoxins, genetic polymorphisms, and the effects of inflammation and oxidative stress [2,3].

Research has demonstrated that valvular surgery, whether performed independently or in conjunction with coronary artery bypass grafting (CABG), is more closely linked to AKI than CABG conducted alone [4,5]. For example, Kim et al. recently reported an AKI incidence of 31.2% among patients undergoing heart valve surgery [6]. Furthermore, Hu et al. found that the incidence of AKI was lower in CABG patients (19.0%) compared to those who underwent on-pump valve surgery (27.5%) or aortic surgery (29.0%) [5].

When considering the most severe form of AKI, Han et al. identified a discharge mortality rate of 54.4% [7], while Skarupskienė et al. documented a concerning mortality rate of 76.5% for AKI patients who needed CRRT [8]. Najjaret et al. reported a much lower in-hospital mortality rate within the same category of patients [9].

The criteria for diagnosing and staging AKI, based on serum creatinine levels and urine output, were established by the Kidney Disease Improving Global Outcomes (KDIGO) AKI work group [10]. However, histological signs of AKI may present earlier than the clinical diagnosis can be made, due to the slow increase of serum creatinine to reach the AKI diagnosis [11]. On the other hand, Lu et al. showed that, in the SAVR setting, even a small increase in postoperative serum creatinine was associated with an increased risk of adverse outcomes [12]. That is why early identification of patients at risk for developing AKI is crucial. Moreover, postoperative AKI, particularly its severe form, can lead to chronic kidney disease progression [3,13], increased hospitalization costs [1,3], and a higher risk of mortality [1,3,6,9,13]. While many biomarkers have been discovered and tested for this purpose, their use in clinical practice remains limited due to financial costs and the lack of standardized application (including timing of measurement, selected cut-off values, and laboratory techniques) [14]. Additionally, the recent 33rd Acute Disease Quality Initiative meeting consensus recommended incorporating sex and gender into AKI research, starting from the fact that the effects of sex on organ crosstalk and the occurrence of AKI are not yet fully understood [15].

Therefore, our retrospective study mainly aims to identify the risk factors and predictors of severe AKI, AKI 3, in patients who undergo on-pump SAVR. Our goal is to help clinicians in the early identification of high-risk AKI patients through the use of cost-effective indicators. This strategy will enhance preoperative risk stratification and facilitate personalized monitoring, allowing for the timely implementation of both preventive and therapeutic interventions.

We analyzed the preoperative, intraoperative, and early postoperative medical data, including hematological values. Furthermore, we examined whether preoperative and early postoperative inflammatory indexes possess significant predictive value for AKI 3. In this context, we compared these indexes to established risk scores, such as the Thakar score, EuroSCORE additive, and EuroSCORE II. Additionally, we investigate potential sex differences in the predictive capability of inflammatory indexes within this specific clinical setting.

## 2. Materials and Methods

We identified the patients who underwent on-pump SAVR at our tertiary center, the “Prof. C.C. Iliescu” Emergency Institute for Cardiovascular Diseases, Bucharest, Romania, between 2022 and 2024. We excluded the patients who had emergency surgeries, those with incomplete data, cases of acute endocarditis, Bentall procedures, iterative surgeries, and individuals undergoing chronic hemodialysis. This retrospective study was conducted following the Declaration of Helsinki and approved by the Ethics and Studies Approval Committee of “Prof. C.C. Iliescu” Emergency Institute for Cardiovascular Diseases, Bucharest (No. 12032/14 May 2025).

The predefined study endpoint was the occurrence of the most severe form of AKI (AKI 3) in the postoperative period. AKI diagnosis was based on the serum creatinine levels according to the criteria established by the Kidney Disease Improving Global Outcomes (KDIGO) guidelines [10]. AKI stage 3 was identified when the serum creatinine level was either three times greater than the reference serum creatinine, exceeded 4 mg/dL, or when renal replacement therapy was initiated [10]. First, our research aims to identify AKI 3 risk factors and predictors associated with on-pump SAVR. We divided the study population into two subgroups to achieve our objective: patients without AKI stage 3 (AKI 3_absent) and those with AKI stage 3 (AKI 3_present). Furthermore, we examined the significance of various preoperative and postoperative inflammatory indexes in predicting AKI stage 3. We compared these indexes with established risk scores, including the Thakar score [16], EuroSCORE additive, and EuroSCORE II, across the entire study population as well as separately for male and female patients. Additionally, we aimed to demonstrate any sex differences in the prediction of AKI stage 3 based on these inflammatory indexes and risk scores within this clinical context.

The patients, previously diagnosed with aortic valvular disease (AVD) with surgical indication, underwent a multidisciplinary evaluation to identify any additional medical conditions. The preoperative assessment included transthoracic echocardiography, transesophageal echocardiography, coronary angiography, arterial Doppler studies, blood tests, and screening for viral and bacterial infections. All patients undergoing SAVR received general anesthesia, underwent full sternotomy, and were placed on normothermic non-pulsatile CPB. Standard and invasive monitoring techniques, including intraoperative transesophageal echocardiography, were utilized. After surgery, patients were closely monitored and cared for in the intensive care unit (ICU) for optimal recovery.

Preoperative, intraoperative, and early postoperative medical data were collected. We defined the early postoperative period as the time immediately following ICU admission. We studied the following hematologic data: leukocyte (L), neutrophils (N), lymphocytes (Lf), monocytes (M), platelet (P) count, Red Cell Distribution Width-Standard Deviation (RDW-SD), Platelet Distribution Width (PDW), and mean platelet volume (MPV). These parameters were reviewed before surgery and during the early postoperative period. We also retrospectively calculated some inflammatory indexes in these two perioperative moments and the postoperative to preoperative delta values as well. Neutrophil-to-lymphocyte ratio (NLR), platelet-to-lymphocyte ratio (PLR), monocytes-to-lymphocyte ratio (MLR), aggregate index of systemic inflammation (AISI), systemic inflammatory response index (SIRI), and systemic inflammatory index (SII) were assessed using the following formulas: NLR = N/Lf; PLR = P/Lf; MLR = M/Lf; AISI = N × P × M/Lf; SIRI = N × M/Lf; SII = N × P/Lf.

Biological sex, age, preoperative creatinine clearance (Clear_preop_creat), body mass index (BMI), AVD type, presence of bicuspid valve, preoperative left ventricle ejection fraction (LVEF), preoperative atrial fibrillation (Preop AF), preoperative hemoglobin concentration (Hb_preop), and risk scores, the Thakar score, additive EuroSCORE and EuroSCORE II, were also reviewed as preoperative variables. During the intraoperative phase, we reviewed the type of surgery performed (either single SAVR or a more complex procedure), the type of prosthesis used (bioprosthesis or mechanical), the length of the surgery, as well as the duration of CBP (CBP_time) and aortic cross-clamping (ACC_time). Postoperatively, we reported early postoperative vasoactive-inotropic score (VIS), the need for hemostasis reintervention, and the occurrence of in-hospital death, besides the hematological data.

Our research used SPSS (Version 30) for statistical analysis. We set the threshold for statistical significance at 95% (*p* ≤ 0.05) and reviewed the data in two subgroups: AKI 3_absent and AKI 3_present patients. We displayed our quantitative variables using the median and interquartile range (IQR), as the distribution was not normal (Shapiro–Wilk test). To compare these variables between the two subgroups, we employed the Mann–Whitney test. Categorical variables were reported as counts and percentages (n%) and analyzed using Fisher’s exact test. The Mann–Whitney test was employed to compare the inflammatory index values between female and male patients. This analysis was conducted for the entire study population as well as within the two subgroups. We used box plots to graphically illustrate the inflammatory indexes preoperatively and following admission to the ICU. The Wilcoxon signed-rank test was employed to assess the significance of changes in each variable across the perioperative period.

The univariable binary logistic regression was used to assess the statistical relationship of each tested variable with the predefined endpoint. The statistically significant variables were evaluated for multicollinearity. The variables with a variance inflation factor (VIF) lower than five were included in the statistical model, which was subsequently analyzed using multivariable binary logistic regression. We reported the statistical significance of the model, the percentage of accurately classified cases, and the results of the Hosmer–Lemeshow test. We aimed to identify the independent risk factors for the occurrence of postoperative AKI 3. We provided the odds ratio (OR), 95% confidence interval (CI 95%), and the corresponding *p*-values for our findings.

The receiver operating characteristic curve (ROC) was employed to assess the predictive ability of the quantitative variables to the binary outcome, the AKI 3 occurrence, and to classify them in this setting. We reported the area under the curve (AUC), its significance (*p*-value), and a 95% confidence interval (CI 95%). When the AUC reached statistical significance, we also reported the cut-off values using the Youden index, with their corresponding sensitivity (Ss) and specificity (Sp). We also reported the ROC analysis of the inflammatory indexes and risk scores, to compare them with the occurrence of AKI 3 in female and male patients (AUC, *p*, CI 95%).

## 3. Results

### 3.1. Data Presentation

A total of 616 patients previously diagnosed with AVD underwent SAVR at the “Prof. C.C. Iliescu” Emergency Institute for Cardiovascular Diseases in Bucharest, Romania, between 2022 and 2024. Our retrospective study analyzed a final cohort of 422 patients since we excluded 194 patients from the analysis. A total of 121 patients (28.67%) exhibited some form of postoperative AKI, with 27 (6.39%) patients in the AKI 3_present subgroup. Figure 1 provides the diagrams of the study. Figure 2 presents the distribution of patients from the postoperative AKI point of view.

A great percentage of AKI 3 patients have died during the same hospitalization (17/27, 62.96%). The patients diagnosed with AKI 3 did not present different features when compared to patients without AKI 3 in terms of age, sex, type of AVD, AV bicuspidate, BMI, preoperative hemoglobin concentration, the presence of preoperative atrial fibrillation, or type of prosthesis used (*p* > 0.005, Table 1). The AKI 3_present patients had significantly higher preoperative L (*p* = 0.013), N count (*p* = 0.005), preoperative SIRI (*p* = 0.048) and AISI (*p* = 0.042), and lower preoperative creatine clearance (*p* = 0.007) (Table 1). Postoperative hematologic data revealed that AKI 3 patients had significantly higher RDW-SD (*p* = 0.002), L (*p* = 0.006), N (*p* = 0.014), and Lf counts (*p* = 0.004) while exhibiting lower *p* (*p* = 0.040) counts when compared to those without AKI 3 (Table 1). None of the postoperative inflammatory indexes differed between the studied subgroups, except for postoperative PLR (*p* = 0.001), which was significantly lower in AKI 3_present patients (Table 1). The AKI 3_present subgroup displayed a higher additive EuroSCORE, EuroSCORE II, and Thakar score (*p* = 0.008, *p* = 0.001, respectively, *p* = 0.001), increased intraoperative time (*p* = 0.001), longer CPB (*p* = 0.001), ACC duration (*p* = 0.001), and a significantly greater need for hemostasis reintervention (*p* = 0.001) (Table 1). When looking at the postoperative-preoperative delta values regarding the inflammatory indexes, we found that only delta PLR (*p* = 0.003) presented different values in the two studied subgroups (Table 1).

We displayed the inflammatory indexes using box plot graphs (Figure 3), both preoperatively and following ICU admission across the two patient subgroups being studied.

The postoperative values of the inflammatory indexes were significantly higher when compared to their preoperative values (*p* = 0.001, Wilcoxon signed-rank test for SIRI, AISI, NLR, MLR, SII). PLR presents a different pattern. Its postoperative levels were not significantly different compared to the preoperative setting (*p* = 0.728). The same analysis within the two subgroups revealed that, in the AKI 3_present patients, all inflammatory indexes presented significant changes in postoperative values compared to their preoperative levels (Table 2). A similar pattern was observed in the AKI 3_absent patients, except for PLR, which did not show any significant difference between postoperative and preoperative values (*p* = 0.277, Table 2).

When comparing inflammatory indexes between female and male patients, we observed significant differences regarding the preoperative and postoperative values of SIRI, AISI, and MLR (Table 3). The changes between postoperative and preoperative values (delta values) of SIRI, SII, and PLR also significantly differed in female patients compared to male patients (Table 3). This pattern was consistent across the entire study population and among the AKI 3_absent patients. However, in AKI 3_present patients, none of the studied inflammatory indexes showed significant differences between female and male patients (Table 3).

### 3.2. The Binary Logistic Regression Analysis

We introduced each of our tested variables in univariable binary logistic regression (Table 4).

From the preoperative variables, only preoperative L (OR 1.200; CI 95%: 1.015–1.419; *p* = 0.033) and N count (OR 1.269; CI 95%: 1.039–1.551; *p* = 0.020), preoperative SII (OR 1.001; CI 95%: 1–1.002, *p* = 0.027), and EuroSCORE II (OR 1.576; CI 95%: 1.285–1.933; *p* = 0.001) presented significant results in univariable analysis (Table 4). We report the same results for the surgical complexity (OR 3.146; CI 95%: 1.301–7.608; *p* = 0.011), CBP_time (OR 1.026; CI 95%: 1.017–1.036; *p* = 0.001), ACC_time (OR 1.029; CI 95%: 1.017–1.040; *p* = 0.001), and intraoperative time (OR 2.246; CI 95%: 1.662–3.085; *p* = 0.001) (Table 3). From the postoperative tested variables, VIS (OR 1.065; CI 95%: 1.040–1.091; *p* = 0.001), postoperative L (OR: 1.094; CI 95%: 1.030–1.161; *p* = 0.003), N (OR 1.090; CI 95%: 1.021–1.164; *p* = 0.009)), Lf count (OR 2.313; CI 95%: 1.444–3.704; *p* = 0.001), RDW-SD (OR 1.089; CI 95%: 1.014–1.170; *p* = 0.019), and hemostasis reintervention (OR 11.297; CI 95%: 4.869–26.214; *p* = 0.001) showed significant results in univariable analysis (Table 4). None of the postoperative inflammatory indexes showed significant results in this analysis. However, the SII postoperative-preoperative delta value (OR 1.006; CI 95%: 1–1.011; *p* = 0.043) and PLR postoperative-preoperative delta value emerged with significant results (OR 1.001; CI 95%: 1–1.002; *p* = 0.028), with an inverse correlation relationship with the endpoint (Table 4).

The variables with significant results in univariable binary logistic regression for severe AKI occurrence were further tested for multicollinearity. The variables with VIF less than 5 were introduced in our model to be tested in multivariable analysis. This model was significant (χ (8) = 66.17, *p* = 0.001), classified 94.1% of cases, with a Nagelkerke R Square of 0.383, and a nonsignificant result in the Hosmer–Lemeshow test (χ (8) = 9.235, *p* = 0.323).

The multivariable analysis revealed that hemostasis reintervention (OR 9.76, CI 95%: 3.565–26.716, *p* = 0.001), VIS (OR1.049, CI 95%: 1.013–1.086, *p* = 0.007), postoperative Lf (OR 2.252, CI 95%: 1.224–4.144, *p* = 0.009), and SII_0-SII_Preop (OR 1.001, CI 95%: 1–1.001, *p* = 0.018) presented statistically significant results (Table 4). These variables showed a direct relationship with our endpoint, except for SII_0-SII_Preop. Upon closer examination of each variable, we noted that in the SII_0-SII_Preop case, the 95% confidence interval includes 1, and its odds ratio is much closer to 1 as well. Therefore, we concluded that the only independent risk factors for AKI 3 in open-pump SAVR, as identified in our study, were hemostasis reintervention, VIS at ICU admission, and postoperative lymphocyte count.

### 3.3. ROC Analysis (AKI 3 Endpoint)

From the tested preoperative variables, we found that only N count (AUC 0.663; CI 95%: 0.562–0.765; *p* = 0.002), L count (AUC 0.643; CI 95%: 0.530–0.755; *p* = 0.013), AISI (AUC 0.617; CI 95%: 0.504–0.730; *p* = 0.043), and SIRI (AUC 0.614; CI 95%: 0.504–0.724; *p* = 0.043), along with EuroSCORE II, Thakar score and additive EuroSCORE, predicted AKI 3 occurrence in our on-pump SAVR patients (Table 5). Among all preoperative variables, EuroSCORE II (AUC 0.758; CI 95%: 0.662–0.854; *p* = 0.001) was a better predictor of the endpoint than the preoperative hematological variables, followed by the Thakar score (AUC 0.678; CI 95%: 0.578–0.778; *p* = 0.001) (Table 5). Regarding EuroSCORE additive (AUC 0.649; *p* = 0.001; CI 95%: 0.540-0.759), only N_Preop, from the preoperative variables, exceeded this risk score in predicting AKI 3 occurrence (Table 5). The intraoperative time also succeeded a good prediction of the endpoint in ROC analysis, with AUC 0.778 (CI 95%: 0.684–0.872; *p* = 0.001), along with the CBP (AUC 0.752; CI 95%: 0.632–0.872; *p* = 0.001) and ACC (AUC 0.728; CI 95%: 0.608–0.848; *p* = 0.001) duration (Table 5).

The early postoperative VIS showed the highest AUC in our ROC analysis of AKI 3 occurrence, with an AUC of 0.861 (CI 95%: 0.794–0.928; *p* = 0.001) (Table 5). Some postoperative hematological variables predicted the endpoint as well: RDW-SD (AUC 0.675; CI 95%: 0.583–0.767; *p* = 0.001), Lf count (AUC = 0.664; CI 95%: 0.545–0.784; *p* = 0.007), L count (AUC = 0.658; CI 95%: 0.542–0.774; *p* = 0.007), P count (AUC = 0.618; CI 95%: 0.507–0.729; *p* = 0.037) (Table 5). From the postoperative inflammatory indexes, only PLR predicted AKI 3 occurrence (AUC 0.689; CI 95%: 0.566–0.812; *p* = 0.003) with an inverse relationship with the endpoint (Table 5). Additionally, the change in PLR value between postoperative and preoperative levels also served as a predictor for the endpoint (AUC 0.668; CI 95%: 0.566–0.812; *p* = 0.003) (Table 5).

We also studied the predictors of the AKI 3 occurrence separately in female and male sex patients (Table 6). VIS at ICU admission, CPB, aortic cross-clamping duration, and EuroSCORE II presented significant AUC both in female and in male sex patients. The rest of the tested variables presented significant AUC only in female sex patients (Table 6). EuroSCORE additive and some inflammatory indexes, such as preoperative SIRI and AISI, postoperative PLR, and the preoperative to postoperative PLR delta difference, significantly predicted AKI 3 occurrence only in female sex patients (Table 6, Figure 4). Postoperative PLR has an inverse relationship with the endpoint. Regarding the Thakar score, we found out that this risk score predicted severe AKI only in male patients (AUC 0.704, CI 95%: 0.583–0.825, *p* = 0.001), while in female patients, the result was without statistical significance (AUC 0.638, CI 95%: 0.454–0.821; *p* = 0.141). None of the studied inflammatory indexes succeeded in predicting the endpoint in male patients (Table 6, Figure 4).

## 4. Discussion

We demonstrated that the occurrence of the postoperative AKI 3 after on-pump SAVR has as independent risk factors the VIS at ICU admission (OR1.049, CI 95%: 1.013–1.086, *p* = 0.007), the surgical reintervention for hemostasis (OR 9.76, CI 95%: 3.565–26.716, *p* = 0.001), the postoperative Lf count (OR 2.252, CI 95%: 1.224–4.144, *p* = 0.009). In our 422 on-pump SAVR cohort, we found that 121 patients (28.67%) presented a form of postoperative AKI, with 27 patients (6.39%) diagnosed with postoperative AKI 3, the most severe form of AKI, that required CRRT. A high percentage of these AKI 3 patients died (17/27, 62.96%).

This was in line with other studies. A prospective international observational multi-center clinical research, EPIS-AKI, reported an AKI incidence of 25.9% in cardiac surgery, with 8.7% patients requiring renal replacement therapy (RRT) [3,17]. Another research, The Randomized Evaluation of Normal versus Augmented Level Replacement Therapy Study, presented a 62.3% overall mortality in patients with AKI requiring RRT [3,13].

VIS, a bedside score, may serve as an indicator of hemodynamic stability, as it sums up all the vasoactive and inotropic medications used to optimize the patient’s hemodynamic status. Researchers have already explored the role of VIS in predicting mortality and morbidity, although there is no consensus on the appropriate timing for measuring VIS [18,19,20]. Pölkki et al. reported that VIS during the first 24 h after ICU admission was associated with outcomes in both general ICU patients and those admitted following cardiac surgery and compared it to the sequential organ failure assessment in this setting [18]. Hou et al. showed that, in cardiac surgery, the max postoperative VIS was significantly linked to postoperative AKI and the need for RRT in AKI patients [19]. Recently, Sun et al.’s meta-analysis reported that elevated early postoperative VIS was associated with various adverse outcomes, including AKI (OR 1.26, 95% CI 1.13–1.41) [20]. The early postoperative VIS was an independent risk factor of severe AKI in our analysis, succeeding the best prediction of AKI 3 (AUC 0.861, CI 95%: 0.794–0.928, *p* = 0.001), with approximately equal results in female and male sex analysis.

Hemodynamic disturbances, renal hypoperfusion, and venous renal congestion are significant factors contributing to AKI occurrence in cardiac surgery [3,21]. Current AKI management of AKI focuses on risk reduction and prevention [14], although the variability of this condition presents unexpected challenges in finding solutions [22]. The KDIGO guidelines recommend a perioperative bundle of care in high-risk patients to prevent and reduce the AKI severity, which includes optimization of perfusion, pressure, and fluid management, halting nephrotoxic medication, avoiding contrast agents, and close monitoring of postoperative kidney function [14]. Effective perioperative hemodynamic management is crucial in this context, yet there is no consensus on the ideal mean arterial pressure during CPB. Intraoperative hypotension during cardiac surgery is linked to a higher AKI risk and can even lead to severe forms of AKI [21,23,24]. On the other hand, the recently issued CPB guideline stated that targeting higher blood pressure led to no difference in AKI occurrence [21]. The guideline highlights the more critical actions during CPB, which include optimizing oxygen delivery (DO2), maintaining adequate CPB blood flow, and monitoring tissue perfusion markers (near-infrared spectroscopy, superior vena cava oxygen saturation), rather than increasing arterial blood pressure with vasopressors [21]. Additionally, Ranucci et al. emphasized the concept of the “optimal” perfusion pressure during CPB within the context of personalized medicine [25]. Other studies focused on the postoperative target arterial pressure in the AKI setting. Smith et al. reported that hypotension frequently occurs within the first 12 h following cardiac surgery [26], prolonged episodes of hypotension being strongly associated with AKI development [26]. Additionally, He et al. found that a postoperative mean arterial pressure of less than 75 mmHg for one hour or more increases the AKI in cardiac surgery [27]. the variability of the pathology.

The surgical reintervention for controlling hemostasis was an AKI 3 independent risk factor in our present study. Ruel et al. also demonstrated the significant link between the surgical reexploration for bleeding with postoperative renal insufficiency, among other complications [28]. On the other hand, Vlasov et al. reported that, in cardiac surgery, it is not the perioperative bleeding itself that predicts AKI, but rather the associated hypotension, increased fluid balance, and the need for transfusions [29]. This finding highlights the need to prevent postoperative bleeding to prevent AKI. Additionally, a newly issued consensus statement emphasized that timely control and management of bleeding is crucial for improving patient outcomes in cardiac surgery [30].

The postoperative lymphocyte count was another independent risk predictor of AKI 3 in our study. The perioperative change of lymphocytes in cardiac surgery has not been extensively studied in the literature about AKI. Instead, research has primarily focused on inflammatory indexes involving lymphocytes in the AKI setting in cardiac surgery. Shi et al. demonstrated that both surgical stress and CPB can induce PBL apoptosis, which may lead to lymphopenia after open heart surgery with CPB [31]. Jiménez-Aguilar et al. showed that the lymphocyte count decrease may be related to the apoptosis of cytotoxic T lymphocytes in children who undergo cardiac surgery with CPB, and to the apoptosis of B lymphocytes in children not receiving CPB [32], suggesting that CPB is not the main cause of this decrease [32]. In cardiac surgery, the lymphocyte count fall may suggest excessive decline in immune system capabilities and immunological dysfunction [33]. Li et al. proposed that monitoring T cell lymphopenia is a better way to predict early postoperative infection compared to C-reactive protein and procalcitonin [34]. Accordingly, Guo et al. found that patients with sepsis-induced AKI had T lymphopenia, with reduced peripheral CD3+ and CD3 + CD8+ T-lymphocyte [35]. Additionally, Cao et al. demonstrated that lymphocytes mediated non-autoimmune AKI and the subsequent chronic kidney disease [36].

Regarding the inflammatory indexes involving lymphocytes, NLR was the most studied in the cardiac surgery setting. Wheatley et al.’s systematic review and meta-analysis reported a statistically significant association between elevated preoperative NLR and postoperative AKI (OR 1.45, 95% CI 1.18–1.77), as well as postoperative need for RRT (OR 2.37, 95% CI 1.50–3.72) in various cardiac surgeries [37]. Studying only on-pump CABG patients, Guangqing et al. showed that patients with higher NLR values presented a higher risk of postoperative AKI when compared to patients with lower NLR values [38]. Postoperative NLR (OR 1.17, 95% confidence interval 1.11–1.23; *p* < 0.001) was an AKI independent predictor of AKI in the Parlar et al. study [39].

The SII, a more complex index calculated by multiplying the NLR by the platelet count, was also examined in the AKI context. SII may serve as a predictor for the risks associated with coagulation and inflammation [33]. The death of lymphocytes contributes to endothelial dysfunction and can lead to abnormal aggregation and thrombosis following platelet activation [33]. In a study by Griffin et al. on on-pump cardiac surgery, postoperative thrombocytopenia was inversely associated with AKI 3, with an odds ratio of 0.966 (95% CI, 0.951–0.982; *p* < 0.001) [40]. Additionally, Li et al. discovered that patients with elevated postoperative SII values experienced higher rates of severe postoperative AKI [41]. Jia et al. presented SII as a novel and easily accessible biomarker for predicting the prognosis of AKI patients [42]. Meanwhile, Aykut et al. discovered a J-shaped relationship between SII and all-cause mortality in critically ill patients with AKI [43]. We did not identify preoperative or postoperative SII values as risk factors for AKI 3. Instead, we found that the change in SII from preoperative to postoperative measurements yielded a statistically significant result in multivariable logistic regression. However, the odds ratio (OR) was close to 1, and the 95% confidence interval (CI) included 1. Therefore, we cannot consider this variable an independent risk factor for AKI 3 in our cohort of SAVR patients. These findings should be validated in a larger, more robust prospective study.

The ROC analysis revealed that early postoperative VIS, intraoperative time, EuroSCORE II, CPB, and ACC time significantly predicted the AKI 3 occurrence both in female and male sex patients with AUC > 0.7 (Table 6). The additive EuroSCORE had a significant, although low, AUC only in female patients (Table 6), while the Thakar score succeeded in predicting AKI 3 only in male patients. Several other studies also studied EuroSCORE variants in predicting AKI, including AKI 3 [24,44,45,46]. Among the preoperative tested inflammatory indexes, only preoperative SIRI (AUC 0.700, *p* = 0.019) and AISI (AUC 0.712, *p* = 0.011) predicted the occurrence of AKI 3 in SAVR female patients, better than the additive EuroSCORE (AUC 0.692, *p* = 0.011) but less accurate compared to EuroSCORE II (AUC 0.841, *p* = 0.001). Postoperative PLR and preoperative-to postoperative PLR delta value presented the same result, although there was an inverse relationship between postoperative PLR and AKI 3 occurrence. We must underscore the fact that in male SAVR patients, none of the studied inflammatory indices, nor the additive EuroSCORE, predicted our endpoint. Shvartz et al. also studied inflammatory indexes in patients with severe aortic stenosis who underwent on-pump SAVR, finding no significant results in the AKI setting [47]. Anyhow, we did not find any research studying the sex difference in AKI 3 prediction starting from the inflammatory indexes. Squiccimarro et al. recently reported that female sex was independently associated with postoperative systemic inflammatory response syndrome (SIRS) and poorer outcomes in cardiac surgery [48]. Dysregulated CD39 and CD73 in female compared to male patients may drive fibrosis and inflammation via eATP. Additionally, females show higher levels of pro-inflammatory markers such as MDC, GM-CSF, ENA-78, and leptin compared to males, indicating a more pronounced inflammatory response [49]. Differences in baseline physiology and responses to injury—such as variations in vascular function, inflammatory responses, antioxidant levels, and other protective pathways—might help explain the sex differences observed in AKI [15].

Our study has some limitations, mainly due to its retrospective and single-center nature. Our cohort is heterogeneous, although all the patients underwent on-pump cardiac surgery involving SAVR. Additionally, we lacked sufficient data on protein C-reactive levels, other indicators of inflammatory status, and details regarding transfusion and cardioplegic solutions. This limited our ability to perform a more comprehensive analysis. Notably, the assessment of AKI 3 relied exclusively on creatinine measurements, omitting information regarding urine output. We also restricted our comparison of inflammatory indexes to the EuroSCORE II, additive EuroSCORE, and Thakar score due to inconsistencies in the available data for other scoring systems. These limitations should be taken into consideration when interpreting our findings, as they may affect the overall validity and applicability of our results.

We successfully identified cost-effective factors that contribute to assessing the risk of AKI 3 in patients undergoing on-pump SAVR. These factors included routine blood analysis (specifically postoperative lymphocytes), the VIS at ICU admission, and the need for hemostatic reintervention. This method enhances preoperative risk assessment and enables personalized monitoring for patients. Our analysis of the EuroSCORE II, additive EuroSCORE, and Thakar score showed different performance results based on sex. Furthermore, we found the inflammatory indexes’ role in this setting only in female sex patients. Anyhow, we did not find any other research studying the sex difference in AKI 3 prediction in cardiac surgery starting from the inflammatory indexes. Further prospective research is needed to identify cost-effective elements from routine blood analyses that could improve risk stratification, taking into account sex and gender in AKI research. Moreover, studies should evaluate the relationship between inflammatory markers, AKI prediction, preoperative medication, and intraoperative anesthetic approaches.

## 5. Conclusions

EuroSCORE II demonstrated significant predictive ability for AKI 3 in both male and female patients undergoing SAVR. The additive EuroSCORE proved effective only for female patients, while the Thakar score applied solely to male patients. Notably, inflammatory indexes were significant predictors only in females. The VIS score, assessed at ICU admission, emerged as an independent risk factor and the strongest predictor of AKI 3. Other independent risk factors included the need for surgical hemostasis control and the postoperative lymphocyte count. These findings highlight the value of effective and cost-efficient tools for identifying patients at risk for severe AKI, thereby enhancing preoperative risk stratification and enabling personalized monitoring.

## Figures and Tables

**Figure 1 diagnostics-15-02211-f001:**
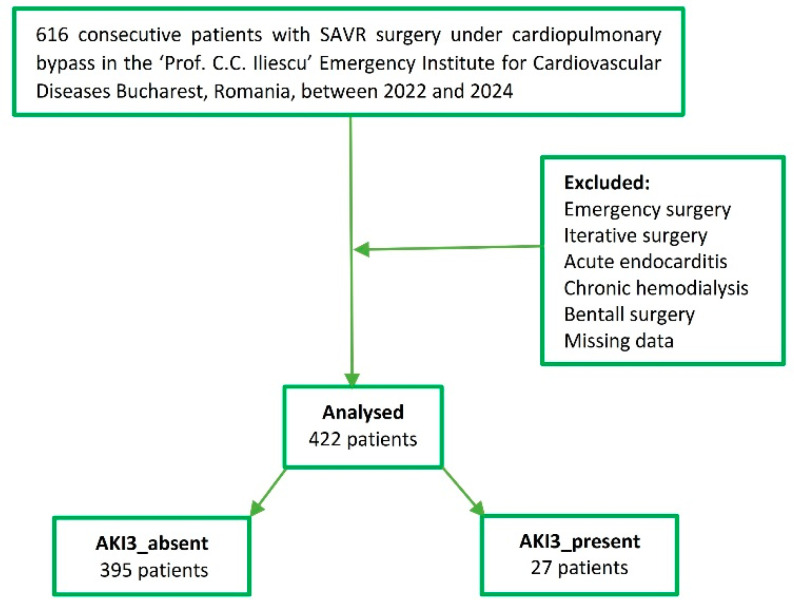
The diagram of the study.

**Figure 2 diagnostics-15-02211-f002:**
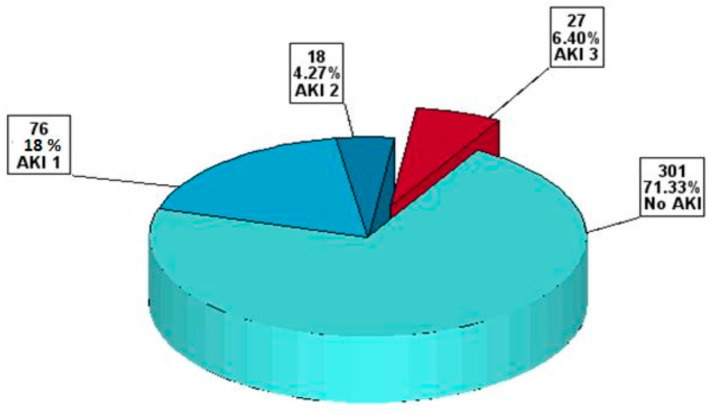
The distribution of the study population from the postoperative AKI point of view. Abbreviation: AKI, acute kidney injury; AKI 1–3, AKI stage 1–3.

**Figure 3 diagnostics-15-02211-f003:**
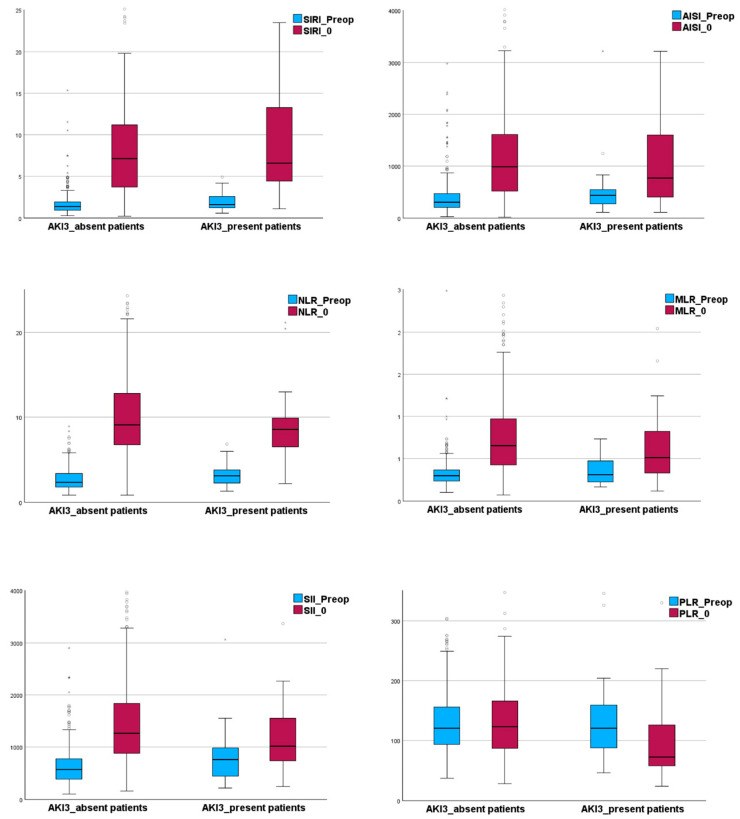
The box plot graphs showing the studied inflammatory indexes preoperatively and following ICU admission in AKI 3_absent and AKI 3_present patients. Note that: “◦”represents mild outliers; “*” represents extreme outliers. Abbreviation: AISI, aggregate index of systemic inflammation; AKI, acute kidney injury; MLR, monocytes to lymphocyte ratio; NLR, neutrophils to lymphocyte ratio; PLR, platelet to lymphocyte ratio; Postop, postoperative; Preop, preoperative; SII, systemic inflammatory index; SIRI, systemic inflammatory response index.

**Figure 4 diagnostics-15-02211-f004:**
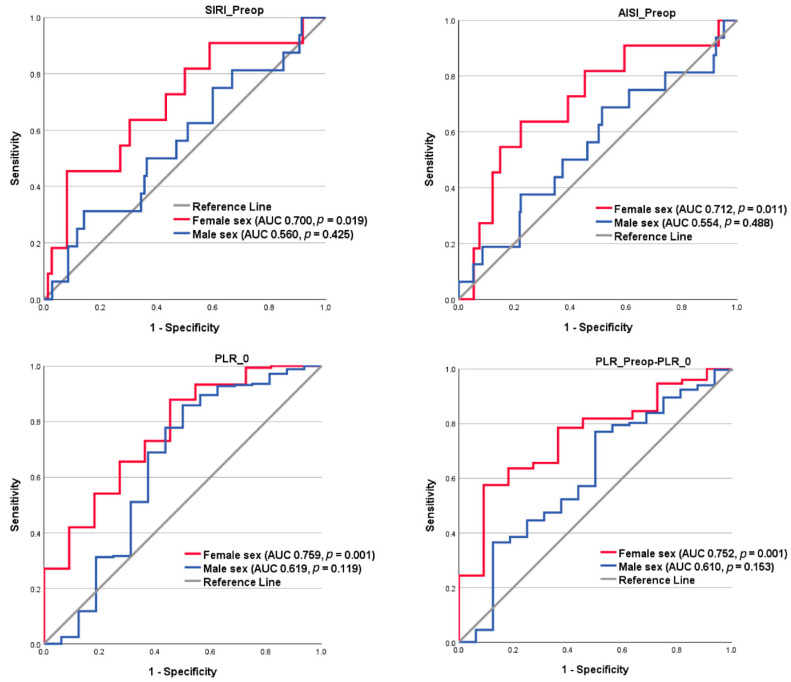
ROC display of the inflammatory indexes with significant results in female and male sex patients studying the AKI 3 endpoint. Abbreviations: AISI_Preop, preoperative aggregate index of systemic inflammation; AKI, acute kidney injury; AUC, area under the curve; PLR_Preop, preoperative platelet-to-lymphocyte ratio; PLR_0, postoperative platelet-to-lymphocyte ratio.

**Table 1 diagnostics-15-02211-t001:** The medical data of the patients reviewed in the two subgroups: AKI 3_absent and AKI 3_present subgroups.

Variable	AKI 3_Absent (*n* = 395)	AKI 3_Present (*n* = 27)	*p* ^1^
Age (years) ^2^	66 [58–70]	64 [57–69]	0.904
Male sex ^3^	247 (62.53%)	16 (59.25%)	0.838
Severe aortic stenosis ^3^	331 (83.79%)	21 (84%)	0.423
Preop AF ^3^	37 (9.36%)	4 (14.81%)	0.318
Hemostasis reintervention ^3^	30 (7.59%)	13 (48.14%)	0.001
Bicuspid aortic valve ^3^	118 (29.87%)	8 (29.62%)	1
Complex surgery ^3^	188 (47.59%)	20 (74.07%)	0.009
Bioprosthetic valve ^3^	264 (66.83%)	18 (66.67%)	1
CBP_time (min) ^2^	95 [80–118]	162 [99–181]	0.001
ACC_time (min) ^2^	72 [60–90]	120 [74–132]	0.001
Clear_preop_creat (mL/min) ^2^	87 [69–112.08]	79.33 [52.94–98]	0.007
BMI (kg/m^2^) ^2^	28 [24.34–32.11]	29.5 [22.3–33.1]	0.462
VIS ^2^	4 [0–8.4]	22 [9.5–30]	0.001
EuroSCORE ^3^	6 [4–7]	7 [5–9]	0.008
EuroSCORE II ^3^	1.5 [1.07–2.42]	3.13 [2.2–4.15]	0.001
Hb_preop (g/dL) ^2^	13.5 [12.4–14.6]	13.2 [11.3–14.2]	0.153
Thakar score ^3^	2 [2–3]	3 [2–4]	0.001
RDW-SD_Preop (fL) ^2^	42.7 [40.5–45.10]	43.4 [41.60–45.80]	0.220
PDW_Preop (fL) ^2^	12.90 [11.70–14.40]	13.3 [12.10–14.40]	0.556
MPV_Preop (fL) ^2^	10.8 [10.2–11.5]	10.9 [10.3–11.60]	0.466
L_Preop (*10^3^/μL) ^2^	7.40 [6.31–8.82]	8.84 [6.91–9.87]	0.013
N_Preop (*10^3^/μL) ^2^	4.59 [3.72–5.77]	5.78 [4.45–6.53]	0.005
M_Preop (*10^3^/μL) ^2^	0.58 [0.47–0.73]	0.61 [0.47–0.83]	0.311
P_Preop (*10^3^/μL) ^2^	226 [193–271]	234 [195–291]	0.659
Lf_Preop (*10^3^/μL) ^2^	1.92 [1.48–2.39]	1.99 [1.45–2.59]	0.718
RDW-SD_0 (fL) ^2^	42.3 [39.8–44.6]	44.4 [41.9–46.8]	0.002
PDW_0 (fL) ^2^	12.5 [11.2–14.10]	12.80 [11.5–13.9]	0.539
MPV_0 (fL) ^2^	10.9 [10.2–11.5]	10.9 [10.3–11.7]	0.606
L_0 (*10^3^/μL) ^2^	12.62 [9.71–15.84]	15.90 [11.72–22.04]	0.006
N_0 (*10^3^/μL) ^2^	10.62 [7.81–13.53]	12.92 [9.7–18.05]	0.014
M_0 (*10^3^/μL) ^2^	0.77 [0.51–1.05]	0.89 [0.51–1.16]	0.540
P_0 (*10^3^/μL) ^2^	138 [112–172]	124 [99–135]	0.040
Lf_0 (*10^3^/μL) ^2^	1.13 [0.83–1.58]	1.68 [1.02–2.55]	0.004
SIRI_Preop ^2^	1.37 [0.92–1.95]	1.61 [1.17–2.79]	0.048
AISI_Preop ^2^	301.04 [207.72–473.30]	435.70 [272.43–545.30]	0.042
SII_Preop ^2^	568.08 [383.33–775.84]	761.38 [427.54–992.79]	0.092
NLR_Preop ^2^	2.32 [1.75–3.37]	3.10 [2.18–3.95]	0.073
MLR_Preop ^2^	0.29 [0.23–0.36]	0.31 [0.21–0.48	0.461
PLR_Preop ^2^	120.65 [93.71–156.48]	120.27 [87.22–159.89]	0.964
SIRI_0 ^2^	7.10 [3.72–11.22]	6.57 [4.42–13.63]	0.972
AISI_0 ^2^	989.26 [518.90–1613.20]	768.77 [399.36–1608.91]	0.599
SII_0 ^2^	1266.23 [875.86–1853.29]	1016.67 [736.72–1572]	0.080
NLR_0 ^2^	9.07 [6.72–12.81]	8.54 [6–10.05]	0.328
MLR_0 ^2^	0.65 [0.42–0.97]	0.50 [0.31–0.87]	0.212
PLR_0 ^2^	122.98 [86.67–166.37]	72.67 [57.44–128.43]	0.001
SIRI_ Preop-SIRI_0 ^2^	8.62 [5.04–13.08]	8.67 [5.92–15.47]	0.756
AISI _0-AISI_Preop ^2^	524.61 [38.93–1297]	655.92 [−130.71–1740.53]	0.516
SII_ Preop -SII_0 ^2^	525.95 [339.93–726.37]	715.38 [380.74–942.39]	0.096
NLR_0-NLR_Preop ^2^	6.72 [4.25–10.15]	5.20 [3.72–7.36]	0.099
PLR_0-PLR_Preop ^2^	3.49 [−30.06–36.93]	−28.72 [−63.71–11.05]	0.003
MLR_0-MLR_Preop ^2^	0.33 [0.11–0.65]	0.12 [−0.04–0.60]	0.112
Intraop_time (hours) ^2^	5 [4–5]	6 [5–7]	0.001
Death ^3^	0 (0%)	17 (62.96%)	0.001

^1^ *p*-value of Mann–Whitney test for quantitative variables or exact Fisher test for categorical variables. ^2^ data are presented as median [IQR]; ^3^ data are presented as n (%). Note that: “*” is a multiplication sign. Abbreviations: ACC_time, aortic cross clamping time; AISI_Preop, preoperative aggregate index of systemic inflammation; AISI_0, postoperative aggregate index of systemic inflammation; AV, aortic valve; AKI, acute kidney injury; BMI, body mass index; CI, confidence interval; CPB, cardiopulmonary bypass; Clear_preop_creat, preoperative creatinine clearance; Hb_preop, preoperative hemoglobin concentration; Intraop_time, duration of the surgery; L_Preop, preoperative leukocyte count; L_0, postoperative leukocyte count; M_Preop, preoperative monocytes count; MLR_Preop, preoperative monocyte-to-lymphocyte ratio; MLR_0, postoperative monocyte-to-lymphocyte ratio; Lf_Preop, preoperative lymphocyte count; Lf_0, postoperative lymphocyte count; LVEF, ejection fraction of left ventricle; M_0, postoperative monocyte count; MPV_Preop, preoperative Mean Platelet Volume; MPV_0, postoperative Mean Platelet Volume; NLR_Preop, preoperative neutrophil-to-lymphocyte ratio; NLR_0, postoperative neutrophil-to-lymphocyte ratio; N_Preop, preoperative neutrophil count; N_0, postoperative neutrophil count; *p* probability value; P_Preop, preoperative platelet count; P_0, postoperative platelet count; PDW_Preop, preoperative Platelet Distribution Width; PDW-0, postoperative Platelet Distribution Width; PLR_Preop, preoperative platelet to lymphocyte ratio; PLR_0, postoperative platelet-to-lymphocyte ratio; Preop AF, preoperative atrial fibrillation; RDW-SD_Preop, preoperative Red blood cell Distribution Width-standard deviation; RDW-SD_0, postoperative Red blood cell Distribution Width-standard deviation; SII_Preop, preoperative systemic inflammatory index; SII_0, postoperative systemic inflammatory index; SIRI_Preop, preoperative systemic inflammatory response index; SIRI_0, postoperative systemic inflammatory response index; VIS, early postoperative vasoactive-inotropic score.

**Table 2 diagnostics-15-02211-t002:** The statistical significance of the preoperative to postoperative trend of the inflammatory indexes.

Variable	AKI 3_Absent (*p*) ^1^	AKI 3_Present (*p*) ^1^
SII	0.001	0.001
SIRI	0.001	0.001
AISI	0.001	0.001
NLR	0.001	0.001
MLR	0.001	0.001
PLR	0.011	0.277

^1^ *p*-value of the Wilcoxon signed-rank test significance. Abbreviation: AISI, aggregate index of systemic inflammation; AKI, acute kidney injury; MLR, monocytes to lymphocyte ratio; NLR, neutrophils to lymphocyte ratio; PLR, platelet to lymphocyte ratio; SII, systemic inflammatory index; SIRI, systemic inflammatory response index.

**Table 3 diagnostics-15-02211-t003:** The Mann–Whitney test significance when studying the inflammatory indexes differences in female and male patients in the whole study population and within the two subgroups.

Variable	Study Population (*p* ^1^) (*n* = 422)	AKI 3_Absent (*p* ^1^) (*n* = 395)	AKI 3_Present (*p* ^1^) (*n* = 27)
SIRI_Preop	0.001	0.001	1
AISI_Preop	0.004	0.001	0.544
SII_Preop	0.756	0.948	0.481
NLR_Preop	0.551	0.507	0.827
MLR_Preop	0.001	0.001	1
PLR_Preop	0.053	0.074	0.422
SIRI_0	0.001	0.001	0.645
AISI_0	0.001	0.001	0.342
SII_0	0.080	0.103	0.512
NLR_0	0.441	0.464	0.680
MLR_0	0.001	0.001	0.318
PLR_0	0.085	0.148	0.318
SIRI_Preop-SIRI_0	0.001	0.001	0.512
AISI_Preop-SIRI_0	0.771	0.899	0.577
SII_Preop-SII_0	0.759	0.946	0.481
NLR_Preop-NLR_0	0.490	0.559	0.716
MLR_Preop-MLR_0	0.001	0.001	0.512
PLR_Preop-PLR_0	0.001	0.003	0.050

^1^ *p*-value of Mann–Whitney test. Abbreviations: AISI_Preop, preoperative aggregate index of systemic inflammation; AISI_0, postoperative aggregate index of systemic inflammation; AKI, acute kidney injury; CI, confidence interval; MLR_Preop, preoperative monocyte-to-lymphocyte ratio; MLR_0, postoperative monocyte-to-lymphocyte ratio; NLR_Preop, preoperative neutrophil-to-lymphocyte ratio; NLR_0, postoperative neutrophil-to-lymphocyte ratio; *p* probability value; PLR_Preop, preoperative platelet-to-lymphocyte ratio; PLR_0, postoperative platelet-to-lymphocyte ratio; SII_Preop, preoperative systemic inflammatory index; SII_0, postoperative systemic inflammatory index; SIRI_Preop, preoperative systemic inflammatory response index; SIRI_0, postoperative systemic inflammatory response index.

**Table 4 diagnostics-15-02211-t004:** The univariable and multivariable binary logistic regression analysis, studying the AKI 3 occurrence (*n* = 422).

Variable	Univariable			Multivariable	
	Exp(B)	OR (CI 95%)	*p*	OR (CI 95%)	*p*
Age (years)			0.670		
Sex			0.734		
Prothesis type			0.986		
Hemostasis reintervention	11.297	11.297 (4.869–26.214)	0.001	9.76 (3.565–26.716)	0.001
Preop AF			0.360		
Bicuspid AV			0.979		
Complex surgery	3.146	3.146 (1.301–7.608)	0.011		0.906
CBP_time (min)	1.026	1.026 (1.017–1.036)	0.001		
ACC_time (min)	1.029	1.029 (1.017–1.040)	0.001		
Clear_preop_creat (mL/min)			0.070		
BMI (kg/m2)			0.412		
VIS	1.065	1.065 (1.040–1.091)	0.001	1.049 (1.013–1.086)	0.007
EuroSCORE II	1.576	1.576 (1.285–1.933)	0.001		0.511
Hb_preop (g/dL)			0.087		
RDW-SD_Preop (fL)			0.220		
PDW_Preop (fL)			0.538		
MPV_Preop (fL)			0.464		
L_Preop (*10^3^/μL)	1.200	1.200 (1.015–1.419)	0.033		
N_Preop (*10^3^/μL)	1.268	1.269 (1.039–1.551)	0.020		
M_Preop (*10^3^/μL)			0.427		
P_Preop (*10^3^/μL)			0.101		
Lf_Preop (*10^3^/μL)			0.689		
RDW-SD_0 (fL)	1.089	1.089 (1.014–1.170)	0.019		0.772
PDW_0 (fL)			0.635		
MPV_0 (fL)			0.616		
L_0 (*10^3^/μL)	1.094	1.094 (1.030–1.161)	0.003		
N_0 (*10^3^/μL)	1.090	1.090 (1.021–1.164)	0.009		
M_0 (*10^3^/μL)			0.206		
P_0 (*10^3^/μL)			0.144		
Lf_0 (*10^3^/μL)	2.313	2.313 (1.444–3.704)	0.001	2.252 (1.224–4.144)	0.009
SIRI_Preop			0.248		
AISI_Preop			0.057		
SII_Preop	1.001	1.001 (1–1.002)	0.027		
NLR_Preop			0.100		
MLR_Preop			0.716		
PLR_Preop			0.221		
SIRI_0			0.512		
AISI_0			0.870		
SII_0			0.888		
NLR_0			0.421		
MLR_0			0.880		
PLR_0			0.348		
SIRI_0-SIRI_Preop			0.434		
AISI _0-AISI_Preop			0.062		
NLR_0-NLR_Preop			0.601		
PLR_ 0-PLR_ Preop	0.994	1.006 (1–1.011)	0.043		0.710
MLR_0-MLR_Preop			0.767		
SII_0-SII_Preop	0.999	1.001 (1–1.002)	0.028	1.001 (1–1.001)	0.018
Intraop_time (hours)	2.246	2.246 (1.662–3.085)	0.001		

Note that: “*” is a multiplication sign. Abbreviations: ACC_time, aortic cross clamping time; AISI_Preop, preoperative aggregate index of systemic inflammation; AISI_0, postoperative aggregate index of systemic inflammation; AV, aortic valve; AKI, acute kidney injury; BMI, body mass index; CI, confidence interval; CPB, cardiopulmonary bypass; Clear_preop_creat, preoperative creatinine clearance; Hb_preop, preoperative hemoglobin concentration; Intraop_time, duration of the surgery; L_Preop, preoperative leukocyte count; L_0, postoperative leukocyte count; M_Preop, preoperative monocyte count; MLR_Preop, preoperative monocyte-to-lymphocyte ratio; MLR_0, postoperative monocyte-to-lymphocyte ratio; Lf_Preop, preoperative lymphocyte count; Lf_0, postoperative lymphocyte count; LVEF, ejection fraction of left ventricle M_0, postoperative monocyte count; MPV_Preop, preoperative Mean Platelet Volume; MPV_0, postoperative Mean Platelet Volume; NLR_Preop, preoperative neutrophil-to-lymphocyte ratio; NLR_0, postoperative neutrophil-to-lymphocyte ratio; N_Preop, preoperative neutrophil count; N_0, postoperative neutrophil count; OR, odds ratio; *p* probability value; P_Preop, preoperative platelet count; P_0, postoperative platelet count; PDW_Preop, preoperative Platelet Distribution Width; PDW-0, postoperative Platelet Distribution Width; PLR_Preop, preoperative platelet-to-lymphocyte ratio; PLR_0, postoperative platelet-to-lymphocyte ratio; Preop AF, preoperative atrial fibrillation; RDW-SD_Preop, preoperative Red blood cell Distribution Width-standard deviation; RDW-SD_0, postoperative Red blood cell Distribution Width-standard deviation; SII_Preop, preoperative systemic inflammatory index; SII_0, postoperative systemic inflammatory index; SIRI_Preop, preoperative systemic inflammatory response index; SIRI_0, postoperative systemic inflammatory response index; VIS, early postoperative vasoactive-inotropic score.

**Table 5 diagnostics-15-02211-t005:** The results of the ROC analysis studying AKI 3 occurrence as a binary endpoint (*n* = 422).

Variable	ROC			Cut Off		
	AUC	*p*	CI 95%	Value	Ss (%)	Sp (%)
VIS	0.861	0.001	0.794–0.928	7.9	88.9	72.2
Intraop_time (hours)	0.778	0.001	0.684–0.872	5.5	70.4	75.9
EuroSCORE II	0.758	0.001	0.662–0.854	2.19	77.8	70.6
CBP_time (min)	0.752	0.001	0.632–0.872	158.5	59.3	91.9
ACC_time (min)	0.728	0.001	0.608–0.848	109.5	63	86.3
PLR_0	0.689	0.003	0.566–0.812	72.9	85.8	51.9
Thakar score	0.678	0.001	0.578–0.778	2.5	74.1	53.7
RDW-SD_0 (fL)	0.675	0.001	0.583–0.767	44.15	59.3	71.1
PLR_0-PLR_Preop	0.668	0.003	0.559–0.777	22.33	63	70.6
Lf_0 (*10^3^/μL)	0.664	0.007	0.545–0.784	1.63	55.6	77
N_Preop (*10^3^/μL)	0.663	0.002	0.562–0.765	5.56	55.6	72.7
L_0 (*10^3^/μL)	0.658	0.007	0.542–0.774	14.71	63	66.1
EuroSCORE	0.649	0.001	0.540–0.759	6.5	55.6	67.6
L_Preop (*10^3^/μL)	0.643	0.013	0.530–0.755	8.77	59.3	73.9
N_0 (*10^3^/μL)	0.641	0.020	0.522–0.760	12.23	66.7	64.3
P_0 (*10^3^/μL)	0.618	0.037	0.507–0.729	135.5	53.7	77.8
AISI_Preop	0.617	0.043	0.504–0.730	500.59	44.8	78.5
SIRI_Preop	0.614	0.043	0.504–0.724	2.36	37	84.1
Age (years)		0.903				
Clear_preop_creat (mL/min)		0.129				
BMI (kg/m2)		0.512				
Hb_preop (g/dL)		0.179				
RDW-SD_Preop (fL)		0.188				
PDW_Preop (fL)		0.524				
MPV_Preop (fL)		0.422				
M_Preop (*10^3^/μL)		0.327				
P_Preop (*10^3^/μL)		0.688				
Lf_Preop (*10^3^/μL)		0.728				
PDW_0 (fL)		0.482				
MPV_0 (fL)		0.575				
M_0 (*10^3^/μL)		0.556				
SII_Preop		0.122				
NLR_Preop		0.063				
MLR_Preop		0.515				
PLR_Preop		0.968				
SIRI_0		0.974				
SII_0		0.094				
AISI_0		0.621				
NLR_0		0.295				
MLR_0		0.254				
SIRI_0-SIRI_Preop		0.762				
AISI _0-AISI_Preop		0.547				
NLR_0-NLR_Preop		0.072				
MLR_0-MLR_Preop		0.151				
SII_0-SII_Preop		0.128				

Note that: “*” is a multiplication sign. Abbreviations: ACC_time, aortic cross clamping time; AISI_Preop, preoperative aggregate index of systemic inflammation; AISI_0, postoperative aggregate index of systemic inflammation; AUC, area under the curve; BMI, body mass index; CI, confidence interval; CPB, cardiopulmonary bypass; Clear_preop_creat, preoperative creatinine clearance; Hb_preop, preoperative hemoglobin concentration; Intraop_time, duration of the surgery; L_Preop, preoperative leukocyte count; L_0, postoperative leukocyte count; M_Preop, preoperative monocyte count; MLR_Preop, preoperative monocyte-to-lymphocyte ratio; MLR_0, postoperative monocyte-to-lymphocyte ratio; Lf_Preop, preoperative lymphocyte count; Lf_0, postoperative lymphocyte count; LVEF, ejection fraction of left ventricle; M_0, postoperative monocyte count; MPV_Preop, preoperative Mean Platelet Volume; MPV_0, postoperative Mean Platelet Volume; NLR_Preop, preoperative neutrophil-to-lymphocyte ratio; NLR_0, postoperative neutrophil-to-lymphocyte ratio; N_Preop, preoperative neutrophil count; N_0, postoperative neutrophil count; *p* probability value; P_Preop, preoperative platelet count; P_0, postoperative platelet count; PDW_Preop, preoperative Platelet Distribution Width; PDW-0, postoperative Platelet Distribution Width; PLR_Preop, preoperative platelet-to-lymphocyte ratio; PLR_0, postoperative platelet-to-lymphocyte ratio; RDW-SD_Preop, preoperative Red blood cell Distribution Width-standard deviation; RDW-SD_0, postoperative Red blood cell Distribution Width-standard deviation; ROC, receiver operator characteristic curve; SII_Preop, preoperative systemic inflammatory index; SII_0, postoperative systemic inflammatory index; SIRI_Preop, preoperative systemic inflammatory response index; SIRI_0, postoperative systemic inflammatory response index; Ss, sensitivity; Sp, specificity; VIS, early postoperative vasoactive-inotropic score.

**Table 6 diagnostics-15-02211-t006:** The ROC analysis studying AKI 3 occurrence in female and male sex patients.

Variable	Female Sex Patients			Male Sex Patients		
	AUC	*p*	CI 95%	AUC	*p*	CI 95%
VIS	0.860	0.001	0.755–0.965	0.859	0.001	0.770–0.948
Intraop_time (hours)	0.852	0.001	0.747–0.957	0.727	0.001	0.588–0.865
EuroSCORE II	0.841	0.001	0.732–0.951	0.708	0.001	0.571–0.845
CBP_time	0.777	0.001	0.612–0.941	0.744	0.001	0.578–0.909
PLR_0	0.759	0.001	0.610–0.909		0.119	
PLR_PreopPLR_0	0.752	0.001	0.621–0.883		0.153	
ACC_time	0.724	0.008	0.559–0.889	0.741	0.004	0.579–0.902
AISI_Preop	0.712	0.011	0.548–0.875		0.488	
SIRI_Preop	0.700	0.019	0.533–0.867		0.425	
EuroSCORE	0.692	0.011	0.544–0.841		0.118	
Thakar score		0.141		0.704	0.001	0.583–0.825
SII_Preop		0.090			0.499	
NLR_Preop		0.393			0.522	
MLR_Preop		0.399			0.835	
PLR_Preop		0.847			0.767	
SIRI_0		0.818			0.921	
AISI_0		0.744			0.807	
SII_0		0.182			0.327	
NLR_0		0.393			0.522	
MLR_0		0.503			0.445	
SIRI_Preop-SIRI_0		0.557			0.978	
AISI_Preop-SIRI_0		0.434			0.886	
SII_Preop-SII_0		0.093			0.504	
NLR_Preop-NLR_0		0.119			0.267	
MLR_Preop-MLR_0		0.181			0.468	

The variables are sorted in descending order by the AUC value in female patients. Abbreviations: ACC_time, aortic cross clamping time; AISI_Preop, preoperative aggregate index of systemic inflammation; AISI_0, postoperative aggregate index of systemic inflammation; AUC, area under the curve; CI, confidence interval; CPB, cardiopulmonary bypass; Intraop_time, duration of the surgery; MLR_Preop, preoperative monocyte-to-lymphocyte ratio; MLR_0, postoperative monocyte-to-lymphocyte ratio; NLR_Preop, preoperative neutrophil-to-lymphocyte ratio; NLR_0, postoperative neutrophil-to-lymphocyte ratio; PLR_Preop, preoperative platelet-to-lymphocyte ratio; PLR_0, postoperative platelet-to-lymphocyte ratio; ROC, receiver operator characteristic curve; SII_Preop, preoperative systemic inflammatory index; SII_0, postoperative systemic inflammatory index; SIRI_Preop, preoperative systemic inflammatory response index; SIRI_0, postoperative systemic inflammatory response index; VIS, early postoperative vasoactive-inotropic score.

## Data Availability

They may be available on request via the corresponding author due to privacy and ethical restrictions.

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
