# Peer review of "New Insights in Assessing AKI 3 Risk Factors and Predictors Associated with On-Pump Surgical Aortic Valve Replacement"

_diagnostics, 2025, doi:10.3390/diagnostics15172211_

Round 1
Reviewer 1 Report
Comments and Suggestions for Authors
I have read with great attention and interest the paper entitled "New insights in assessing AKI 3 risk factors and predictors associated with on-pump surgical aortic valve replacement."
First of all, I would like to congratulate the authors on their extensive clinical experience and the results they have achieved.
Nevertheless, I believe the paper could be significantly improved in two key areas:
-
One group shows an in-hospital mortality rate exceeding 60%, which is unacceptable. Why were those patients deemed eligible for surgery? Why was TAVI not considered instead?
-
The level of English is somewhat low, and the entire manuscript should be revised to improve clarity and readability.
The level of English is somewhat low, and the entire manuscript should be revised to improve clarity and readability.
Reviewer 2 Report
Comments and Suggestions for Authors
Thank you for inviting me to complete the review. Akute kidney injury is still a challenging issue in cardiac surgery. Many attempts have been made to address this issue and to come up with a perfect risk score or preventive measures.
Several concerns arise while reading the manuscript:
1). As this was a retrospective analysis, how was post hoc power analysis performed?
2). The patients included in this retrospective study appear to be quite heterogeneous in terms of such important parameters as Hemostasis reintervention, CBP time, complexity of the procedure etc - please see table 1. So it gives me concerns whether it is completely appropriate to make conclusions if we compare such cohorts. It would be great if authors could provide information whether they attempted to perform pseudorandomization to address this issue.
3). The authors are quite vague when writing the Conclusion: "Some hematological data and early postoperative VIS can effectively facilitate the assessment of AKI3 risk in patients..." It is appears very important to be precise in giving conclusions.
4). Did the authors discuss/evaluate the results of their study with general or specific for cardiac surgery risk scores/risk factors for AKI? Such as older risk scores like CICSS, Cleveland, STS (Mehta), SRI, MCSPI, AKICS, or newer risk scores such as AKI-RiSc, The Acute Renal Failure after Cardiac Surgery (Thakar Score) etc.
5). The authors should make a clear emphasis on what's new in their findings, and how it may affect clinical practice or science.
Reviewer 3 Report
Comments and Suggestions for Authors
- Clarify the novelty of your study: what gap does it fill? For example, is it the analysis of inflammatory indices stratified by sex? Or the comparison with EuroSCORE?
- State the time points of the “early postoperative” lab collection more precisely (24h after surgery? ICU admission?).
- Discuss the clinical implications of your findings: how could these predictors be used in practice? Preoperative risk stratification? Personalized monitoring?
- In the discussion session, between lines 410 and 436, it would be useful to underline the variability of the pathology, citing some studies that document AKI in cardiovascular patients with particular and anomalous resolutions (it is recommended to cite this article DOI: 10.24969/hvt.2023.382)
- A limitations paragraph is currently missing or underdeveloped.
Round 2
Reviewer 1 Report
Comments and Suggestions for Authors
No further comments
Reviewer 2 Report
Comments and Suggestions for Authors
The Authors have made substantial improvements.
I believe that the Manuscript can be Published after editorial check.